# Design of buried charged networks in artificial proteins

Mona Baumgart[1], Michael Röpke [1,3], Max E. Mühlbauer [1,3], Sam Asami [1], Sophie L. Mader [1], Kai Fredriksson[1], Michael Groll [1], Ana P. Gamiz-Hernandez [1,2] & Ville R. I. Kaila [1,2 ✉]

Soluble proteins are universally packed with a hydrophobic core and a polar surface that drive the protein folding process. Yet charged networks within the central protein core are often indispensable for the biological function. Here, we show that natural buried ion-pairs are stabilised by amphiphilic residues that electrostatically shield the charged motif from its surroundings to gain structural stability. To explore this effect, we build artificial proteins with buried ion-pairs by combining directed computational design and biophysical experiments. Our findings illustrate how perturbation in charged networks can introduce structural rearrangements to compensate for desolvation effects. We validate the physical principles by resolving high-resolution atomic structures of the artificial proteins that are resistant towards unfolding at extreme temperatures and harsh chemical conditions. Our findings provide a molecular understanding of functional charged networks and how point mutations may alter the protein's conformational landscape.

---

[1] Center for Integrated Protein Science Munich (CIPSM) at the Department Chemie, Technische Universität München, Lichtenbergstraße 4, 85748 Garching, Germany. [2] Department of Biochemistry and Biophysics, Stockholm University, 10691 Stockholm, Sweden. [3] These authors contributed equally: Michael Röpke, Max E. Mühlbauer. ✉email: ville.kaila@dbb.su.se

Protein structures are universally folded by shielding hydrophobic amino acids from the aqueous environment to reach a global free energy minimum[1]. However, buried charged networks provide central functional elements responsible for the biological activity in many proteins[2–4]. In complex I, a redox-driven proton pump essential for cell respiration, a 200 Å network of buried ion-pairs and charged residues provide functional elements responsible for the enzyme's proton pumping machinery[2,5]. Similarly, the heat shock protein 90, an essential molecular chaperone in the eukaryotic cell, employs an extended network of charged amino acids to regulate the enzymatic activity and the chaperone function[3,6].

Charged residues buried within the low-dielectric protein core ($\varepsilon_{\mathrm{protein}} = 4$–$10$) are thermodynamically destabilised relative to the high-dielectric aqueous environment ($\varepsilon_{\mathrm{aq}} = 80$) by the (Born) desolvation free energy associated with moving charged groups towards lower dielectric surroundings. This effect can in part be compensated by the electrostatic interactions between oppositely charged residues, but recent studies[4,7] suggest that each ion-pair destabilises the overall native protein fold by up to 5 kcal mol$^{-1}$. This raises the important question on how proteins with large extended networks with hundreds of charged residues[2,5] are structurally and thermodynamically stabilised. To address this puzzling question without the potential bias of natural protein scaffolds, we introduce here buried ion-pairs into the hydrophobic core of artificial helical bundle scaffolds[8,9] by combining directed computational design and molecular simulations with biophysical, spectroscopic, and structural experiments. These experimentally well-behaved, minimal de novo models do not structurally resemble highly intricate natural proteins, but despite their simple structure, four-helical bundles have been used to probe, e.g., oxidoreductase-[9,10], oxygen-binding-[8], light-capturing[11] and ion-transport properties[12]. Here they have been used to deduce interaction principles of buried ion-pairs without the complicated effect of complex natural surroundings based on rather simple biophysical and computational experiments.

## Results and discussion

### Structural stability of buried ion-pairs in artificial bundle proteins.
To study how the protein stability is affected by buried charged residues, we introduced a glutamate (Glu)/lysine (Lys) ion-pair into the hydrophobic protein core of an artificial 4α-helical protein scaffold[10] at positions 49 and 84 (F49E and F84K), where helix 1 forms contacts with helices 2 and 3 (see Fig. 1a, Methods and Supplementary Table 1). Atomistic molecular dynamics (MD) simulations of this *Maquette 1* model suggest that the overall structure remains stable on the simulation timescale (Fig. 1b), and energetically favours the charged (Glu$^-$/Lys$^+$) state over the neutral (Glu$^0$/Lys$^0$) state (Supplementary Fig. 1d). During the MD simulations, a few water molecules transiently interact with the buried charges and partially destabilise the hydrophobic core relative to the *Maquette 1* model with a non-polar core (Fig. 1c). To further characterise the properties of the constructs, we expressed the proteins in *E. coli*, and studied their stability using circular dichroism (CD) spectroscopy and chemical unfolding experiments. The proteins remain α-helical, as indicated by the minima at 208 and 222 nm (Fig. 1e). However, consistent with findings on ion-pairs in natural proteins[4,7], the introduced charges decrease both the melting temperature by $\Delta T_m > 40\,^\circ\mathrm{C}$ and the protein overall stability by $\Delta\Delta G \approx 5$ kcal mol$^{-1}$ relative to our model with an intact hydrophobic core (Fig. 1d–g).

Removal of the Glu or Lys counter charges decreases the protein stability by around 3 kcal mol$^{-1}$ based on unfolding experiments (Supplementary Figs. 1, 2a). However, despite the

well-packed protein models in our MD simulations and a two-state folding transition in the chemical unfolding experiments, our nuclear magnetic resonance (NMR) experiments show heteronuclear single quantum coherence (HSQC) spectra that are not too well-dispersed and contain some overlapping signals (Supplementary Fig. 2f), an effect that may arise from the highly symmetric structure of these models, and/or involvement of possible molten-globule like states[10]. Although helix dipole-sidechain charge interactions may affect the protein stability[13–15], we find here that inverting the polarity of the ion-pair in the *Maquette* 1 model only leads to a minor shift in the melting temperature $(\Delta\Delta T_\mathrm{m} = 0.1\,^\circ\mathrm{C})$ (Supplementary Fig. 1f, 2a). Although we currently lack experimental structural data to draw definite conclusions about the buried ion-pairs in the *Maquette* 1 models, our data support that buried charges decrease the overall protein stability similarly to what has been described for natural proteins[4,7] and consistent with the data for other de novo protein models (see below).

To improve the accuracy of the design, we next employed another artificial protein scaffold (*Maquette 2*) with a resolved core scaffold[11], but a different helical topology (helix 1 interacting with helices 2 and 4) as compared to the *Maquette 1* model (Fig. 1h, inset). Insertion of the ion-pair at positions 17 and 72 (F17E/F72K relative to Maquette *2*/hc) yields a stable structure in the MD simulations (Fig. 1i, Supplementary Fig. 3), but similar to the *Maquette 1* model, the charged element forms transient interactions with a few water molecules during the simulations (Fig. 1j). The ion-pair destabilises the protein by $\Delta\Delta G \approx 2$ kcal mol$^{-1}$ in chemical unfolding experiments (Fig. 1k, n), suggesting an improved stability relative to *Maquette 1*, but an overall destabilisation relative to the construct with the non-polar core. Moreover, the overall protein stability remains unaffected when the charged residues are introduced into the more water accessible region of the bundle (I101E/F50K) (Supplementary Fig. 3c).

### Characterisation of buried ion-pairs.
The *Maquette 2* proteins have well-dispersed HSQC spectra, but the introduced ion-pair leads to significant chemical shift perturbations (Fig. 2c, Supplementary Fig. 2b), indicating possible structural rearrangements to compensate for the electrostatic penalty. The ion-pair is supported by NH$_3$-selective heteronuclear in-phase single quantum coherence spectroscopy (HISQC) measurements at 275 K that reveal slow proton exchange of the shielded sidechains. This NMR signal further disappears upon substitution of either Lys72 or Glu17 with phenylalanine (Fig. 2d), whereas the intensity of the peak indicates dynamic flexibility consistent with our MD simulations (Fig. 2b, Supplementary Fig. 3). Based on 3D-NMR experiments at 293 K, we resolved a structural model of *Maquette* 2 with the introduced charged element (Fig. 2a, Supplementary Fig. 2c, Supplementary Table 2). The model reveals a subtle bending of helices 1 and 3 that could arise from the thermo-dynamic cost associated with the introduced charged element (Figs. 1n and 2a). However, the mixed open/closed ion-pair conformations as indicated by NH$_3$-selective HISQC spectrum (Fig. 2d) and MD simulation data (Fig. 1h, Supplementary Fig. 7b), are not reflected in the model building, which favours dissociation of the ion-pair in the structure calculation (see "Methods" section). Nevertheless, the MD simulations suggest that the closed ion-pair conformation can easily form in the NMR-models by a <1 Å inward motion of helices 1 and 3. These conformational changes are also qualitatively supported by structural modelling (Fig. 2a, Supplementary Fig. 2c). Taken together, our data suggest that the ion-pair introduces some dynamical flexibility and subtle structural rearrangements in the

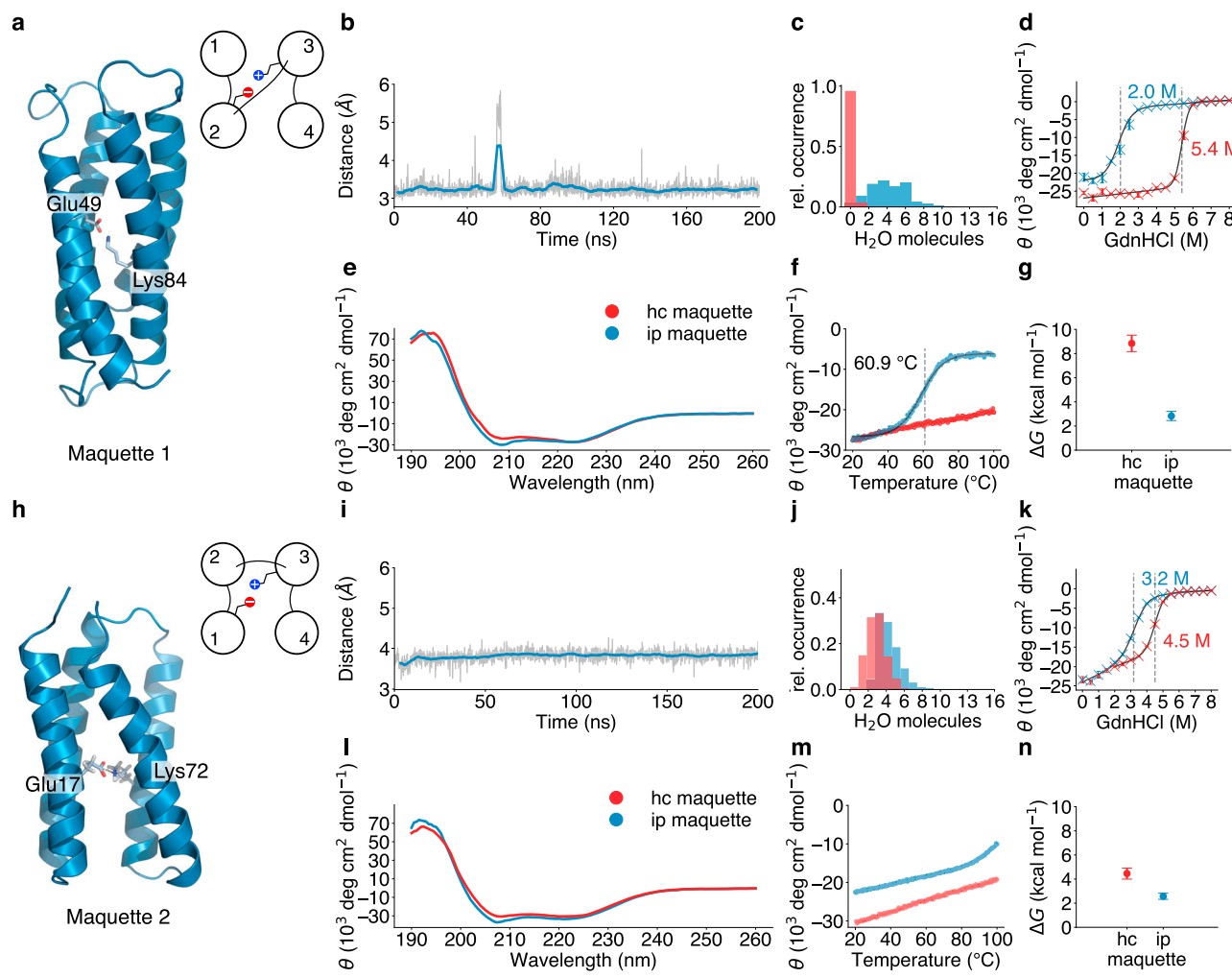

**Fig. 1 Structural stability of buried ion-pairs in artificial bundle proteins. a**, **h** Structural snapshot of protein models; **b**, **i** distances between introduced ion-pairs (the shaded line represents the actual data, the coloured line is a moving average over 50 frames), and **c**, **j** number of buried water molecules during MD simulations within 9 Å of helices 1–4; **e**, **l** CD spectra at 20 °C of protein *Maquettes* 1 and 2 with a hydrophobic core (hc) and introduced ion-pair (ip); **f**, **m** melting curves measured by following the temperature dependence of the 222-nm trough in the CD spectra; **d**, **k** chemical unfolding with GdnHCl up to 8 M monitored by CD spectroscopy. Data points are presented as mean values of triplicates ($n = 3$ independent experiments) with ±standard deviation shown as error bars; and **g**, **n** thermodynamic stability ($\Delta G$) values calculated from the chemical unfolding experiments, with error bars computed from the standard deviation of the chemical unfolding profiles.

bundle scaffold, which might enable exploration of both open and closed ion-pair conformations (Fig. 2a, Supplementary Fig. 3). Such dynamic flexibility coupled to subtle conformational changes are critical for the function of ion-pairs in natural proteins[5,6].

**Analysis and design of buried charged elements in proteins**. To better understand the charge-compensation principles, we next developed a computational design approach, where we directed the residue search based on atomistic molecular simulations, by explicitly modelling both solvation and electrostatic effects underlying the thermodynamic stability of the ion-pairs. To this end, we optimised the Glu17-Lys72 ion-pair (F17E/F72K) by introducing random in silico-point mutations in the protein surroundings, and directed the residue search using a Metropolis Monte Carlo (MC) criterion towards an increased interaction between the ion-pair and the protein structure as an optimisation target. At the same time, we selected for structures with an increased overall protein stability (Fig. 3a, b, see Methods section).

In our directed design, we find that polar and charged residues that form hydrogen-bonded contacts with the buried ion-pair are introduced in 2230 of 2838 (≈80%) designed constructs probed within the first solvation sphere of the charged element (Fig. 3c). In addition to other charged residues, Gln/Asn, Tyr, and Ser/Thr constitute 20% of the sampled substitutions. These mutations insert amphiphilic elements around the charged centre (Fig. 3c) that could enhance the protein stability. In statistical analysis of ca. 180,000 ion-pairs from around 6000 resolved membrane protein structures (Fig. 3d, Supplementary Figs. 4 and 5), we find similar charge-stabilising motifs in natural proteins as predicted in the computational design. Interestingly, we find a 10–30% increased probability of observing charge-stabilising residues (Gln, Asn, Tyr, Ser, Thr) within the first solvation sphere of the ion-pair in the natural proteins (Fig. 3d, Supplementary Figs. 4 and 5), closely resembling the distribution predicted by our directed design (Fig. 3c). Moreover, tyrosine residues that have an overall slightly higher natural abundance, are somewhat over-represented within the membrane proteins dataset (Supplementary Fig. 5b), possibly as they comprise both bulky non-polar/aromatic properties that can be used for packing the protein core or form cation–π interactions with positively charged residues, but also a polar hydroxy headgroup that can form hydrogen

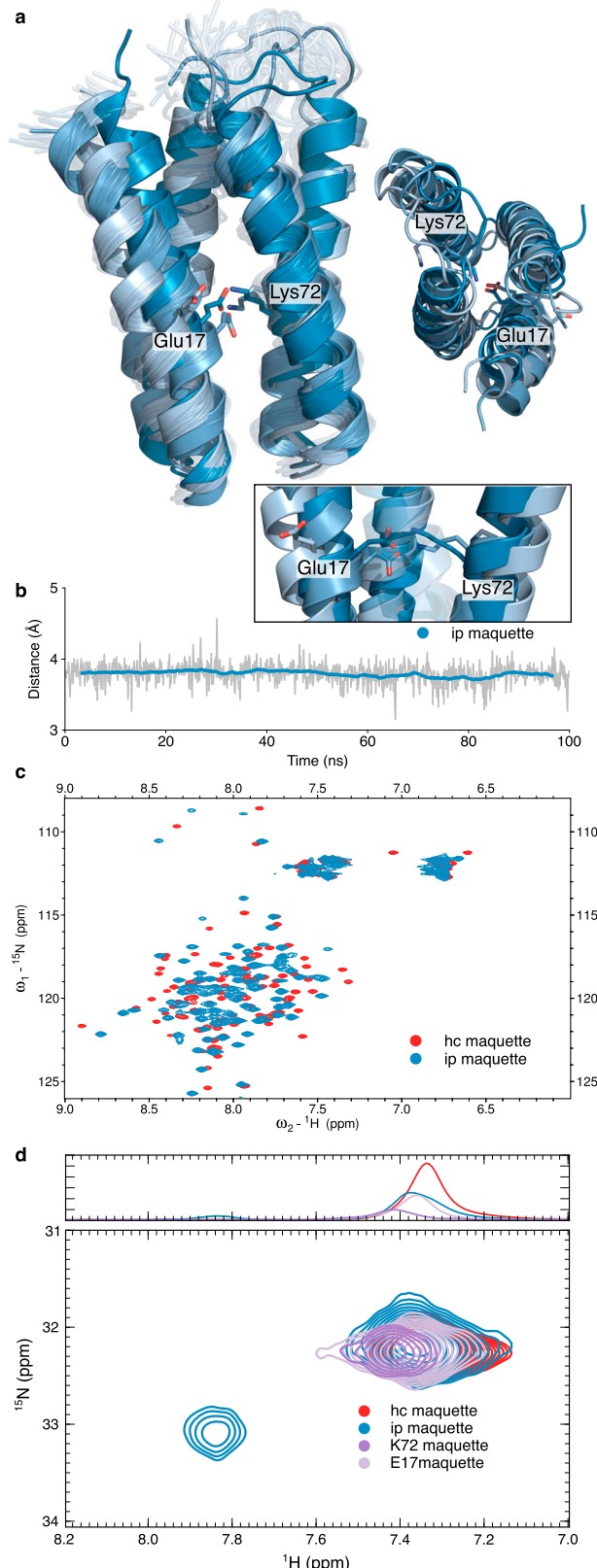

**Fig. 2 Biophysical and structural characterisation of ion-pair in *Maquette* 2.**
**a** NMR-structural model of *Maquette* 2 with an ion-pair at positions 17 and 72 (F17E/F72K) modelled in both closed/open conformations (in dark/pale blue). **b** Dynamics of the Glu17 and Lys72 ion-pair modelled based on the NMR structure. The shaded line represents the actual data, the coloured line is a moving average over 50 frames. **c** Heteronuclear single quantum coherence (HSQC) NMR spectrum of *Maquette* 2 with Glu17-Lys72 ion-pair (ip, F17E/F72K, in blue), and hydrophobic core (hc, in red). **d** $NH_3$-selective HISQC experiment at 275 K, indicative of a protected sidechain with slow proton exchange. Glu17-Lys72 ion-pair (ip, F17E/F72K, in blue), hydrophobic core (hc, in red), and single residue substitutions (K72 *maquette* (E17F) in purple); E17 *maquette* (K72F), in pale purple.

residues[18], are also clearly pronounced in the distribution (Fig. 3d, Supplementary Fig. 4). We note that the buried Trp68 might stabilise the Glu17-Lys72 ion-pair in the *Maquette* 2 model (F17E/F72K, Supplementary Fig. 2c). The ion-pair distributions further suggest that glutamine residues seem to favour Arg-Asp ion-pairs, whilst Lys-Glu ion-pairs favour asparagine residues (Supplementary Fig. 5a), possibly due to steric constraints in natural proteins.

To understand the physical basis of the shielding motifs, we further studied the designed charged elements by electrostatic calculations. For the unshielded ion-pair, the Born desolvation penalty ($\Delta G = +12$ kcal mol$^{-1}$) becomes partially compensated by the electrostatic interaction with the protein surroundings ($\Delta G = -6$ kcal mol$^{-1}$), leading to an overall destabilising effect in the model calculations (Fig. 3e, see Supplementary Information). The partially uncompensated charges expose a non-uniform electric field, which could be responsible for the partial wetting of the protein core during the MD simulations and the overall reduced protein stability (Fig. 1). However, the shielding element provides a clear electrostatic compensation in the model systems that results in enhanced stability (Fig. 3e). The thermodynamic cost of embedding a charge-shielding Gln or Asn residue next to an ion-pair within the protein core is nearly isoenergetic in the model calculations (Supplementary Fig. 2e).

Inspired by these findings, potential charge-stabilising motifs were introduced in the ion-pair surroundings of the *Maquette* 2 (Supplementary Fig. 2d). We find that introduction of Asn at position 69 (Maquette 2/V69N) leads to a small, but statistically significant enhancement of the protein stability in chemical unfolding experiments (Fig. 3f). Moreover, in $NH_3$-selective HISQC measurements, a significant increase in the population of the closed ion-pair is observed, suggesting that the asparagine could stabilise the closed ion-pair conformation (Fig. 3f), as also qualitatively supported by MD simulations (Fig. 3g). The HSQC spectrum of *Maquette* 2/V69N indicates that the protein is well-folded, with a downfield shift of most peaks relative to the ion-pair model (Supplementary Fig. 2g). To further investigate the conformational flexibility of the *Maquette* 2 models, we performed heteronuclear NOE (hetNOE), as well as longitudinal ($T_1$) and transverse ($T_2$) relaxation experiments[19]. The hetNOE and relaxation data of the hydrophobic core, ion-pair, and charge-stabilised ion-pair constructs are very similar (Supplementary Fig. 8), indicating that all models have similar dynamics on the ps-ns timescale, maintain a similar fold on the studied conditions, and are likely to have overall similar structures.

**Rational design of an ultra-stable buried charged network.** To test the physical (Fig. 3d, e) and evolutionary (Fig. 3c) insights, we created a charged network, with a double ion-pair surrounded by four charge-stabilising Gln-motifs embedded

bonds. We note, however, that certain amino acids may have special preferred locations in the protein structure[16]. Moreover, charged elements are pronounced with an 10–40% increased probability[17], whereas hydrophobic residues occur with a 10–60% reduced probability around the ion-pair. Interestingly, tryptophan residues that can form cation-π interactions with, e.g., Lys

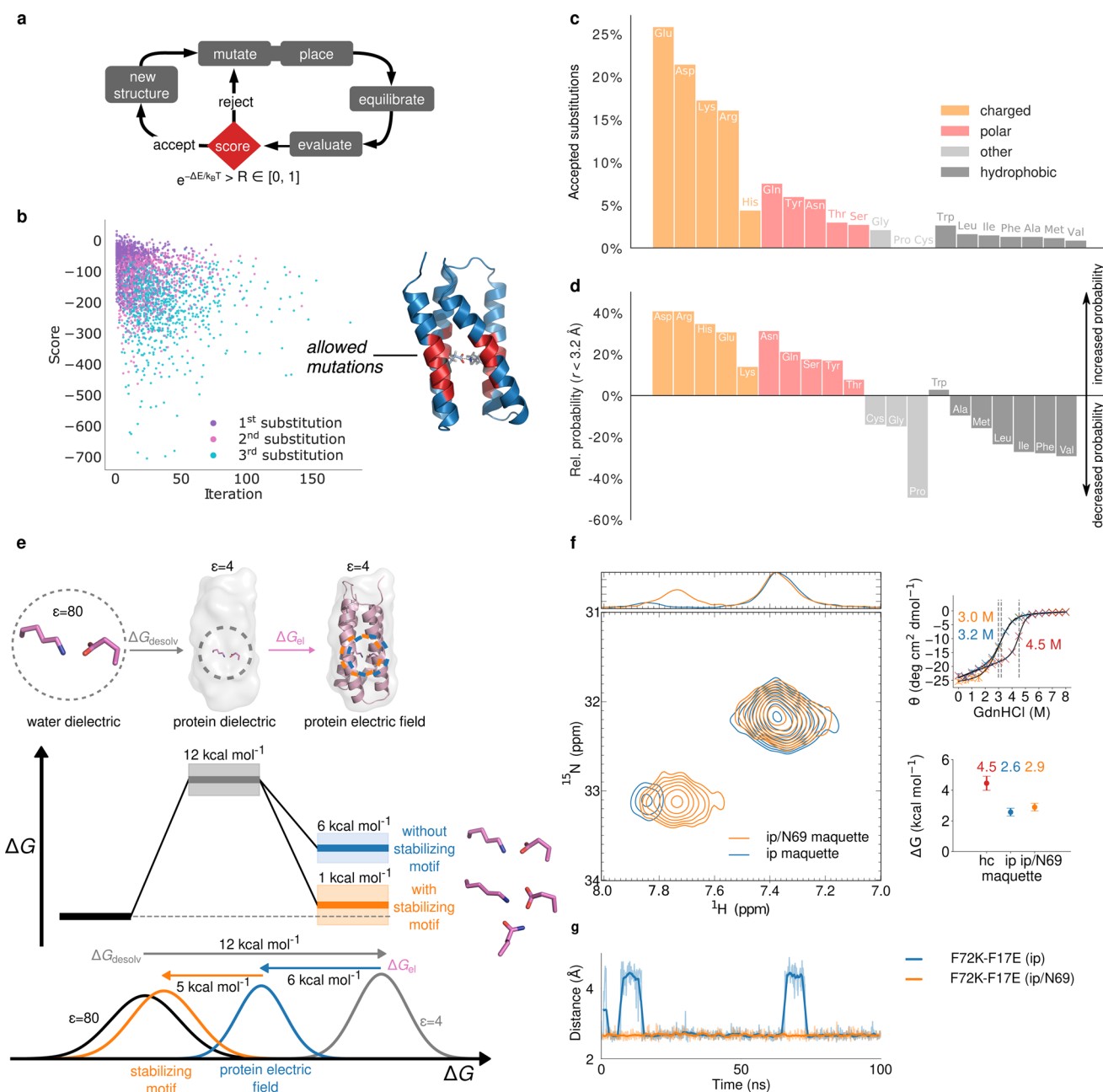

**Fig. 3 Directed design, evolutionary conservation, and biophysical analysis of buried charged elements in proteins. a** Directed computational design algorithm employed to optimise buried charge elements. **b** Sampled energy-score during directed design (see "Methods" section). Inset: *Maquette* 2 model with area of introduced substitutions (in red). **c** Distribution of substituted amino acids in 1000 directed simulations. **d** Relative probability of amino acids surrounding ca. 182,000 natural ion-pairs from ~6000 structures. **e** Electrostatic calculations on the energetics of the ion-pair with and without charge-stabilising motif. **f** NH$_3$-selective HISQC experiment at 275 K of *Maquette* 2 with charge-shielded ion-pair. Right: chemical unfolding with GdnHCl monitored by CD spectroscopy of the *Maquette* 2-protein with stabilising shielding motif (top), and thermodynamic stability ($\Delta G$) values calculated from the chemical unfolding experiments (bottom). Data points of the chemical unfolding profiles are presented as mean values of triplicates ($n = 3$ independent experiments) with ±standard deviation shown as error bars. **g** MD simulations of *Maquette* 2 models at 275 K. The figure shows distances (Glu/OE2-Lys/ NZ) between the ion-pair without (in blue) and with charged-stabilising Asn69 (in orange). The shaded line represents the actual data, and the coloured line is a moving average over 50 frames.

within a ca. 70 Å-long α-helical Maquette-protein framework (*Maquette* 3, Fig. 4c, d, see Extended Methods section), originally a Zn$^{2+}$-site binding scaffold (PDB ID: 5VJT), that provides a larger non-polar interaction surface relative to the minimal 4α-helical *Maquette* models 1 and 2. The designed construct is stable in MD simulations with intact ion-pairs and minimal water penetration (Fig. 4a, Supplementary Fig. 6), and the pronounced

minima at 208 and 222 nm in the CD spectrum support its α-helical secondary structure (Fig. 4c). Remarkably, the *Maquette* 3 model is resistant towards unfolding at >100 °C and 6 M gua-nidine chloride concentrations, indicating a high overall protein stability, despite the large buried charge density. To validate the underlying molecular structure behind the high stability, we resolved the x-ray structure of the *Maquette* 3 model at 1.85 Å

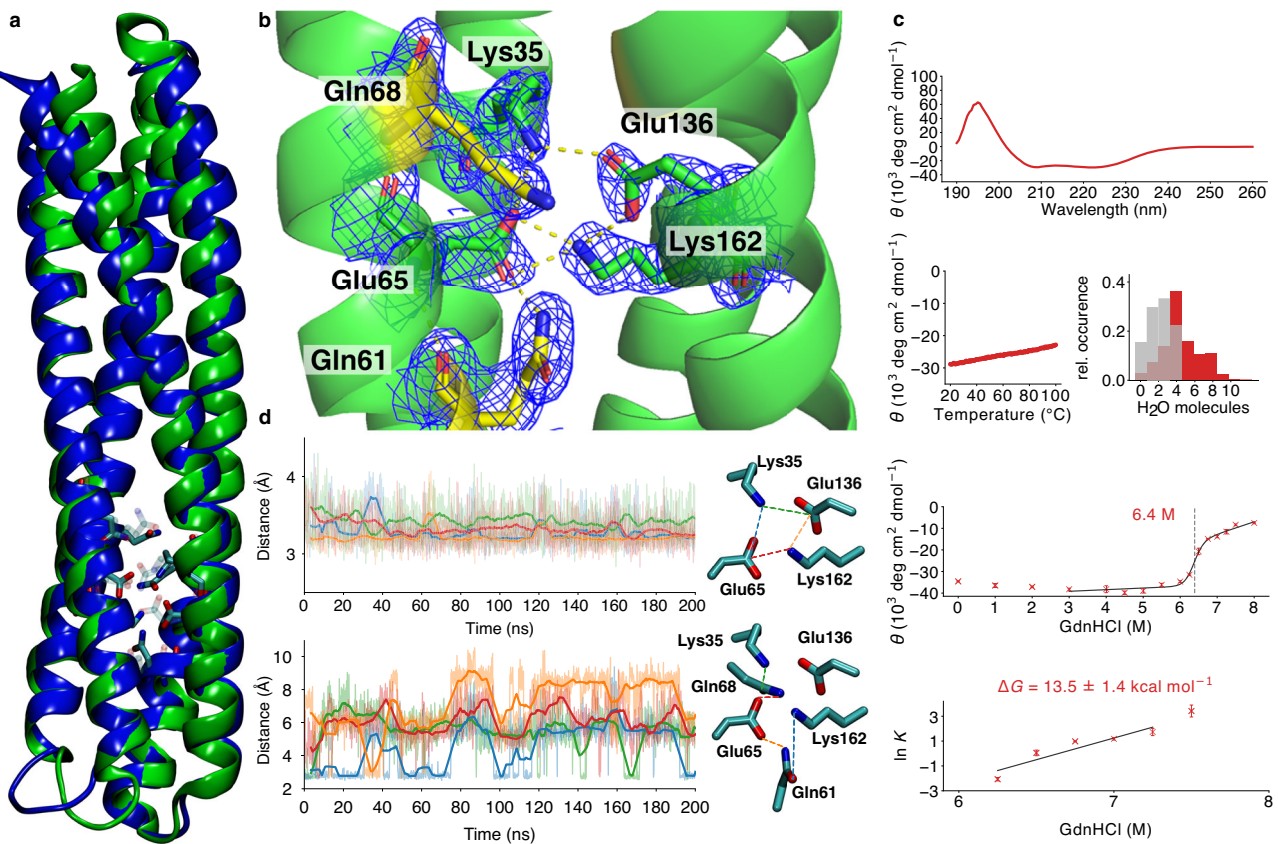

**Fig. 4 Rational design of a buried ultra-stable charged motif. a** Structural snapshots from the in silico model of the *Maquette* 3 protein with a charge-shielded buried ion-pair network (in blue) and the resolved x-ray structure (in green). **b** Closeup of the designed charged network in the 1.85 Å resolved x-ray structure. **c** CD spectrum at 20 °C of *Maquette* 3 (top); melting curves measured by following the temperature dependence of the 222-nm trough from CD spectroscopy (left); count of buried water molecules 5 Å around the ion-pair/Asn surroundings (grey) and water molecules within 5 Å of the complete protein (in red) (right); chemical unfolding with GdnHCl up to 8 M monitored by CD spectroscopy; thermodynamic stability inferred from the chemical unfolding experiments (bottom). Data points of the chemical unfolding profile are presented as mean values of triplicates with ±standard deviation as error bars. **d** MD simulation of the ion-pair dynamics. Top: Distances between Lys35-Glu65 (blue), Lys35-Glu136 (green), Glu65-Lys162 (red), and Glu136-Lys162 (orange). Bottom: Distances between Lys35-Gln68 (green), Gln61-Glu65 (orange), Gln61-Lys162 (blue), and Glu65-Gln68 (red). The measured atomic distances are depicted to the right of the plots. See Supplementary Fig. 6 for further details. The shaded line represents the actual data, the coloured line is a moving average over 50 frames.

resolution (Fig. 4a, b, Supplementary Fig. 6, Supplementary Table 3). The $2F_o$-$F_c$ electron density map closely resembles the designed in silico model (backbone RMSD (in silico/x-ray) = 1.7 Å), with a charged core forming tight ion-pairs between Lys162/Glu65 and Lys35/Glu136 at distances of 2.7–2.9 Å. The charged residues are stabilised by hydrogen bonds to Gln61/68/169/132 (Fig. 4b, d, Supplementary Fig. 6c, e) that dynamically exchange in the MD simulations (Fig. 4d, Supplementary Fig. 7c). Although other charge-stabilising residues, such as tyrosine and histidine, are often present around natural ion-pairs (Supplementary Figs. 4 and 5), these residues were too bulky to be inserted around the charged network in the current minimal *Maquette* 3 model. Intriguingly, the ion-pairs remain tightly sealed from water molecules in the crystal structure, consistent with the high stability indicated by our unfolding experiments and MD simulation (Fig. 4c, Supplementary Fig. 6b). Similar to the *Maquette* 1 models, the HSQC spectrum of *Maquette* 3 (Supplementary Fig. 6f) shows several overlapping and less sharp signals possibly due to the higher internal symmetry of the model. We could therefore not use NMR to study these models further. Nevertheless, to quantify the stabilising effect of the charge-shielding residues surrounding the ion-pairs, we further substituted two of

the glutamines with alanine residues (Q61A, Q68A). Our chemical unfolding measurements suggest that these substitutions indeed destabilise the protein by around 5 kcal mol⁻¹, supporting the proposed charge-shielding effect (Supplementary Fig. 6g). Taken together, the *Maquette* 3 design supports the proposed electrostatic principles in which amphiphilic residues shield charged networks from the non-polar surroundings to gain thermodynamic stability.

In summary, we have characterised here the energetics of buried ion-pairs using artificial protein design in combination with computational, biophysical, and structural experiments. We identified that amphiphilic shielding motifs stabilise the buried charged networks by a fine-tuned balance between desolvation and electrostatic effects. The motifs could represent an evolutionary prerequisite for buried ion-pairs, as they are found in a majority of the analysed natural protein structures, and independently supported by our directed computational design. Our de novo proteins built based on these physical principles, resulted in ultra-stable charged networks that sustain harsh chemical and physical conditions. Our findings may provide a basis for understanding the structural stability of extended charged networks in molecular gates, switches, and allosteric

regulation sites in different protein machineries[5,6,20,21], and how disease-causing mutations may destabilise the protein structure and result in an altered conformational landscape[22–24].

## Methods

**In silico modelling of Maquette models.** An atomic model of *Maquette* 1 was built in silico with α-helical secondary structure restraints placed on the residues 1–27, 36–62, 71–97, and 106–130 based on PsiPred[25]. The helices were packed to maximise the non-polar interaction within the core, in which Phe-15 and Phe-21 were restrained to form a stacking interaction with the nearest anti-symmetric helix, following computational modelling principles applied before[26]. After the initial restrained relaxation, loops were built to connect the helices, followed by restrained minimisation and MD simulations using the CHARMM36 force field[27] at $T = 300$ K and using a 1 fs integration timestep. The *Maquette* 1 models were built and relaxed using CHARMM[28], followed by MD simulations (see below). The *Maquette* 2 models were constructed based on scaffold structures resolved in ref. [11]. The *Maquette* 3 model was built in silico based on a helical coiled-coil template (PDB ID: 5VJT). First, all prosthetic groups were removed from the template, and glycine and alanine residues were replaced by valine and isoleucine residues, respectively, to stepwise maximise the non-polar packing, followed by a short (20 ns) MD simulation to test its stability. To further maximise the non-polar packing, hydrophobic residues at positions H10F, H111F, H125F, H139F, and the charged network was modelled manually by substitutions E162K, E35K, I61Q, H68Q, I132Q, Y169Q according to the charge-shielding principles described in Fig. 3e. The final sequence of the *Maquette* 3 model is shown in Supplementary Table 1.

**Molecular simulations.** Classical MD simulations of the designed proteins were performed by solvating the protein in a TIP3P water box and 150 mM NaCl concentration. MD simulations were performed in duplicates for 200 ns (Supplementary Fig. 7) at $T = 310$ K using a 2 fs integration timestep and the CHARMM36 force field[27], and modelling the long-range electrostatics using the particle-mesh Ewald approach. Solvation free energies were estimated using Poisson-Boltzmann (PB) continuum electrostatic calculations with MC sampling. Protein residues were modelled as atomic point charges, and the protein surroundings was treated as polarisable medium with ε = 4 that was embedded in a water dielectric of ε = 80. The linearized-PB equations were solved numerically using the adaptive PB-solver (APBS) and Karlsberg+[29,30]. Visual Molecular Dynamics (VMD)[31] and PyMOL[32] were used for visualisation and analysis, MD simulations were performed using NAMD2[33]. See Supplementary Information for summary of the performed simulations (Supplementary Fig. 6, Supplementary Table 4).

**Directed computational design.** Charged elements in the artificial protein models were optimised by introducing random mutations into the *Maquette* structures, followed by 10 ps classical MD simulations in explicit solvent. The protein models were scored by calculating the self-interaction of the ion-pair (weight = 0.48) and its interaction with the remaining protein surroundings (weight = 0.48) and all nonbonded interactions within the protein (weight = 0.02). The sum of the weights were normalized to 1.0. The substitutions were accepted or rejected based on weighted interaction scores using a Metropolis Monte Carlo criterion. All substitutions were subjected to 5000 steepest-gradient minimisation steps, followed by 5 ps restrained relaxation with harmonic protein backbone restraints with force constants of 1 kcal mol$^{-1}$ Å$^{-2}$. The protein interactions were modelled using the CHARMM36 force field[27], and the MD relaxation was performed using a 1 fs integration timestep at $T = 310$ K using NAMD2[33]. The directed computational design approach was implemented in Python in our in-house code PACMAN (Packing with MC-assisted networks).

**Statistical analysis of ion-pairs.** Statistical analysis of naturally occurring ion-pairs was carried out based on structures available in the orientations of proteins in membranes (OMP) database[34]. From the 7230 available structures, 6045 with a resolution <3.5 Å were included. The protein surface was calculated using the MSMS program[35], and ion-pairs were identified using the Salt Bridges plugin of VMD[31]. All residues within a 15 Å sphere of the identified ion-pairs were counted and probabilities for their occurrence within a certain distance to the centre-of-mass of the ion-pair was calculated as,

$$p^{aa}(r_{\text{thr}}) = \frac{N_{r_{\text{thr}}}^{aa}}{\sum_i^{\text{AA}} N_{r_{\text{thr}}}^i} \tag{1}$$

where *aa* is the identified amino acid, $r_{\text{thr}}$ is the threshold radius, AA the full set of amino acids, and $N_{r_{\text{thr}}}^{aa}$ is the number of amino acids within a given threshold radius.

**Protein expression and purification.** Genes for the computationally designed proteins were synthesized and cloned into pET21a(+) vector (Genscript). F17E, F72K, 1ip/I43Y, 1ip/I39Q, 1ip/R76S, and 1ip/L91H of *Maquette* 2 were produced using the QuikChange Lightning Site-Directed Mutagenesis Kit (Agilent) according to the manufacturer's manual and the primers listed in Supplementary Table 5.

The C-terminal His$_6$-tagged proteins were overexpressed in *E. coli* BL21 (DE3) for 4 h at 37 °C after induction with 0.5 mM IPTG at OD$_{600}$ 0.7. After sonication, the lysate was applied on a Ni-NTA affinity column. The His-tag was cleaved by a Tobacco Etch Virus protease, and the proteins were purified by size exclusion chromatography using a preparative Superdex 75 pg HiLoad 16/600 column (GE Healthcare) with a flow rate of 1.0 mL min$^{-1}$. See Supplementary Table 1 for amino acid sequences of all artificial proteins and Supplementary Table 6 for DNA sequences. The buffers used during protein purification are summarised in Supplementary Table 7.

**Circular dichroism spectroscopy.** CD spectra and melting curves were measured on a Jasco J-715 system combined with a cooling-heating unit using a 1-mm-path quartz cuvette. Protein concentrations of around 5–10 μM were used for measurements at 20 °C. For the determination of melting curve, samples were heated up from 20 to 100 °C, while the alpha-helical CD-signal at 222 nm was monitored. The protein stability was also studied by guanidine-hydrochloride (GdnHCl) or urea induced unfolding with GdnHCl/urea concentrations up to 8 M. Folding free energies were obtained from linear least-square fits of -$RT$ ln $K$ that were extrapolated to 0 M GdnHCl. The apparent equilibrium constant $K$ was determined from the fraction of the denatured protein, $f_{\text{D}} = (y - y_{\text{N}}) / (y_{\text{D}} - y_{\text{N}})$, where $y$ is the CD raw signal, and $y_{\text{N}}$ and $y_{\text{D}}$ are the signals for the native and denatured states, respectively. Error bars were computed from the standard error of the least-square fits from the average unfolding profiles.

**Nucleic magnetic resonance spectroscopy.** Heteronuclear NMR spectra were recorded using $^{15}$N- and $^{15}$N,$^{13}$C-labelled protein at 20 °C in a buffer with 50 mM sodium phosphate, 100 mM NaCl, and 1 mM EDTA at pH 7.5. NMR experiments were performed on Bruker Avance III spectrometers operating at a $^{1}$H Larmor frequency of 600 MHz (14.09 T) and 950 MHz (22.31 T) using a CPTCI triple-resonance cryoprobe and the Bruker TopSpin v. 3.5 software. Sequence-specific backbone and side-chain assignment were obtained using 3D HNCA, HN(CO)CA, HNCO, HN(CA)CO, HNCACB, CBCA(CO)NH, HBHA(CO)NH, H(CCO)NH, CC(CO)NH, HC(C)H/(H)CCH TOCSY, $^{15}$N/$^{13}$C$^{\text{ali}}$/$^{13}$C$^{\text{aro}}$-edited $^{1}$H,$^{1}$H NOESY (120 ms mixing time) as well as 2D (HB)CB(CGCD)HD. 2D $^{1}$H,$^{15}$N HISQC spectra were recorded for the detection of Lys $^{15}$NH$_3$ groups[36]. $^{1}$H, $^{13}$C, $^{15}$N chemical shifts were referenced via DSS and spectra were assigned using Sparky[37]. $^{15}$N $T_1$, $T_{1\rho}$, and steady-state hetNOE experiments were collected at an external magnetic field of 11.74 T ($^{1}$H Larmor frequency of 500 MHz) and 293 K. Temperature compensation was employed, according to Lakomek et al.[38] The $^{15}$N rf amplitude for the $T_{1\rho}$ spin-lock was set to 1.5 kHz. For the steady-state hetNOE experiment a saturation time of 4 s was employed. $T_2$ was calculated according to

$$\frac{1}{T_2} = \frac{R_{1\rho}}{\sin^2\theta} - \frac{R_1}{\tan^2\theta} \tag{2}$$

with

$$R_{1\rho} = 1/T_{1\rho}, \tag{3}$$

$$R_1 = 1/T_1, \tag{4}$$

$$\tan\theta = \frac{\omega_1}{\Omega}. \tag{5}$$

$\omega_1$ is the amplitude of the $^{15}$N spin-lock field and $\Omega$ is the $^{15}$N resonance offset from the spin-lock carrier frequency. The experimental error was set to two times the standard deviation of the spectral noise. Uncertainties were estimated by 1000 MC runs.

**NMR structure calculation.** Automated NOE assignment was conducted in CYANA[39,40] to generate restraints and calculate de novo structures of the *Maquette* 2 construct with ion-pair at Glu17/Lys72. The closed ion-pair conformation was modelled by restraining lower and upper distance limits between Glu17-OE1 and Lys72-NZ of 2.8 and 3.0 Å, respectively, as guided by the MD simulations. The calculations with torsion angle dynamics were started from 200 random structures, employing 20,000 torsion angle dynamics steps. In addition to distance restraints, dihedral angle restraints derived from chemical shifts were obtained using TALOS-N[41]. The restrained energy minimisation of the consensus bundle of 20 conformers in explicit solvent against the AMBER force field was carried out using OPALp[42,43]. The statistics of the structure calculations is given in the Supplementary Table 2.

**X-ray crystallography.** Crystals of the *Maquette* 3 construct were grown at 20 °C using the sitting drop vapour diffusion method. Drops contained a 1:2 mixture of protein solution (40 mg ml$^{-1}$ protein) and reservoir solution (100 mM HEPES buffer pH 7.5, 1.5 M potassium phosphate). The crystals were cryo-protected with 25% glycerol. Diffraction data were collected at the beamline X06SA at the Paul Scherrer Institute, SLS, Villigen, Switzerland (λ = 1.0 Å). Evaluation of reflection intensities and data reduction were performed with the program package XDS (v. January 31, 2020)[44]. Refinement of the initial model was carried out with the coordinates of the in silico model of *Maquette* 3 using REFMAC5[45], followed by

model building using COOT[46] and CCP4 v. 7.0[47]. See Supplementary Table 3 for further details.

**Reporting summary**. Further information on research design is available in the Nature Research Reporting Summary linked to this article.

## Data availability

The structures have been deposited to the Protein Data Bank, PDB IDs: 6ZOC [rcsb.org/structure/6zoc], 6Z35 [rcsb.org/structure/6z35]. The NMR assignments are deposited in the BMRB under the entry number 34518 [bmrb.io/data_library/summary/?bmrbId=534518]. Other data are available from the corresponding author upon reasonable request. Source data are provided with this paper.

## Code availability

The pacman code is available under MIT license at https://github.com/KailaLab/pacman-protein-design.

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

## Acknowledgements

We acknowledge Prof. Michael Sattler and Prof. Franz Hagn for providing us access to the Bavarian NMR Facility, Astrid König for technical assistance in crystallisation trials, and Dr. Sina Kazemi for helpful discussions regarding NMR structure calculations. This work was supported by the collaborative research centre, SFB1035 (B12) and the Centre for Integrated Protein Science Munich (CIPSM). This work received funding from the European Research Council (ERC) under the European Union's Horizon 2020 research and innovation program/grant agreement no. 715311. VRIK is supported by the Knut and Alice Wallenberg Foundation. Computational resources were provided by the Leibniz-Rechenzentrum (LRZ), SuperMuc (project: pr27xu), and the Swedish National Infrastructure for Computing (SNIC) at PDC partially funded by the Swedish Research Council through grant agreement no. 2018-05973.

## Author contributions

M.B. expressed, purified, and characterised the designed proteins, and performed computational and biophysical characterisation. M.B., M.R. M.E.M., S.L.M., A.P.G.H., and V.R.I.K. performed molecular simulations and design. M.R., M.E.M., S.L.M., and V.R.I.K. developed analytic computational tools. M.B. and S.A. performed NMR experiments and calculated NMR structure ensembles. M.B., M.G., and V.R.I.K resolved the x-ray structures. A.P.G.H. performed initial computational model building. M.B., M.R., M.E.M., S.A., S.L.M., K.F., M.G., A.P.G.H., and V.R.I.K.

analysed results. V.R.I.K. designed and directed the project, performed initial computational design and experimental characterisation, and wrote the manuscript with input from all authors.

## Funding

## Competing interests
The authors declare no competing interests.
