## [Peer Review File · Nature Communications]

REVIEWER COMMENTS

Reviewer #2 (Remarks to the Author):

Notes for review of Buried Charged Residues Nat Comm paper

This paper describes the introduction of ion-pairs into the hydrophobic core of a designed protein as a model system for protein engineering. Overall the work is sound, and adds significantly to the understanding of the effect of buried polar interactions in folded proteins. The paper lacks clarity at times, especially in the (lack of) description of the rationale for the mutations chosen, as well as in some of the methods employed. I believe that this paper should be accepted pending the following alterations.

Specific points:

The abstract refers to a point mutation- this is misleading given that there are two substitutions.

The introduction of the modifications to Maquette 1 is very vague. No mention is made in the text of the specific modifications to this protein sequence. This is clearer for the Maquette 2 modifications, but we still aren't given a clear indication on what has been removed in order to add the ion pair. All protein modifications should be named using the conventional A123B nomenclature, e.g. L50K, where a Leu at position 50 has been changed to a Lys.

Page 2: 'reverting the polarity of the ion-pair' should be 'inverting the polarity of the ion-pair'

Page 2: A 10 degree C change in melting temperature is not a 'small decrease'. This is actually a very large change, and warrants more explanation, especially given the later discussion of the role of peripheral polar residues in stabilising (or not) buried ion pairs.

The discussion of the maquette 3 crystal structure states that the structure 'closely resembles' the models. This should be quantified, either for the snapshot used in the figure, or more meaningfully, for a minimum energy structure arising from the model building process. Either way this should be quantified by an rmsd value (backbone rmsd makes most sense).

Reviewer #3 (Remarks to the Author):

Review

Baumgart et al.

Design of Buried Charged Networks in Artificial Proteins

For Nature Communications, Aug 2020.

This paper describes studies on the design of artificial proteins with charged residue pairs within their cores. The study takes a multi-disciplinary approach to evaluating the resulting model proteins, applying molecular dynamics simulations, structural biology and a range of biophysical techniques. As such, the study is well executed and will no doubt be of interest to the protein design community. However, there are major weaknesses, particularly with the selection of the model systems, that seriously undermine the conclusions. As a result, the paper cannot be published in its present form.

The authors select three model proteins with which to test mutations that introduce ion pairs into the protein core. All three are four-helical bundles and all three stem from earlier efforts by the groups of Dutton and Moser (authors references 8-10) and DeGrado (reference 13) to design proteins that can bind co-factors within the bundle core, with the aim of supporting oxidoreductase

functionalities. This description alone makes the choice of these targets questionable for a study into natural, globular proteins. Little rationale is provided as to why these systems were selected, other than to avoid "the potential bias of natural protein scaffolds". The authors do not clearly establish the relevance of these model systems to proteins such as complex I or Hsp90, which they mention in the introduction.

A case in point is the first example protein, labelled Maquette 1. It is based on a parent designed to bind two heme groups in its core via two bis-histidine ligation sites. The protein consists of four identical repeats of 26 residues connected by glycine/serine linkers. These are intended to fold into a four-helical bundle with an up-down topology. Each repeat has a histidine residue in position 6, such that the bis-histidine sites are formed between helices $\alpha 1$ - $\alpha 3$ at one end of the bundle and helices $\alpha 2$ - $\alpha 4$ at the other. Data presented in reference 10 shows nanomolar affinity for heme, but also that the protein forms a molten globule-like state in the absence of the ligand. NMR data present here in supplementary figure 2f are qualitatively in agreement with that presented in reference 10; i.e. the parent protein lacks a well-defined tertiary structure. Nevertheless, the authors introduce an ion pair into this system, via F49E and F84K mutations, with the aim of introducing an $\alpha 2$ - $\alpha 4$ interaction. The ^{15}N -HSQC spectrum of this Maquette 1 variant, also shown in supplementary figure 2f does nothing to indicate that this fold has been achieved, and is in no way diagnostic of a "well-folded structure", as the authors claim. Given the quality of the NMR data, it is not surprising that details on this variant are not presented - its structure, internal dynamics or the formation of the ion pair - even unique resonance assignment would prove difficult. The Maquette 1 variant is therefore of no value in supporting the author's claims.

The third model protein, Maquette 3, faces similar issues. No parent sequence is provided for this design (e.g. in supplementary Table 1) and its source is not clearly stated. A BLAST search reveals it to be closely related to a protein deposited in the PDB as 5VJS, a longer four-helical bundle designed to accommodate two heme and a Zn^{2+} binding site. From the little information provided, the ion pair introduced into this parent to form Maquette 3 does not appear to be inserted into the hydrophobic core, as the reader may expect, but it overlaps with the Zn^{2+} binding site. The authors do demonstrate that this design has been successful with the crystal structure of a variant where the ion pair has been stabilized by four flanking glutamine residues, which is clearly a publishable result. But these flanking residues do not seem to solely replace hydrophobic residues, but also tyrosine and histidine residues. In the authors' own survey, tyrosine and histidine were identified as more favored flanking residues than the glutamines that replace them (supplementary Fig. 4). This again undermines the main claims of the paper. As no data is presented on the parent, we cannot know how much stabilization has been achieved by the ion pair and flanking mutations, also bearing in mind that the parent design (5VJS) was conceived for functionally binding multiple cofactors, not for stability. As no NMR data is presented on this set of variants, we cannot correlate the prevalence of the ion pair with that reported for Maquette 2. This result may thus be of interest in terms of protein design, but it again does little to provide experimental support for general considerations of how ion pairs are stabilized within the hydrophobic cores of natural proteins.

This leaves us with the example of Maquette 2. Here the parent is the PS1 protein designed by DeGrado (author's reference 13). This follows a similar strategy to the parent proteins of Maquette 1 in that it incorporates a co-factor binding site - in this case for a porphyrin - within a four-helical structure, but aims to support this site with a clearly separated hydrophobic core region. The authors exploit this separation to place an ion pair within the hydrophobic region and seek to stabilize this with various flanking mutations. They then solve the structure of the ion-pair variant by NMR. Although this provides a good example of the applicability of NMR in such cases, there are several questions with the results. Firstly, as the authors mention, the ^{15}N -HSQC fingerprint of the ion-pair variant is surprisingly different to that of the parent protein, indicative of structural changes on a global scale. From Fig. 2c, the overall dispersion of signals for the variant appears smaller than that of the parent and the linewidths of the variant appear larger. It is thus reasonable for the authors to conclude that the variant undergoes considerable internal motion. However, they do not quantify and localize these dynamic changes, for example via NMR relaxation studies. Given the importance of internal dynamics to their argument, this is a puzzling omission. Equally puzzling is the lack of a structure on the V69N stabilized variant. If the flanking mutation is stabilizing the ion pair and decreasing the extent of internal motion, then this should

be evident in the ^{15}N -HSQC spectrum, which should at least partially return to the dispersion and linewidths of the parent. Such data are, however, not provided, nor is any on the five further flanking mutations that were either neutral or mildly destabilizing, according to supplemental Fig. 2d.

Taken together, these issues mean that there is little consistent data that the authors can use to support their general hypothesis. For example, they show that introducing an F49E, F84K ion pair into Maquette 1 results in a ~ 5 kcal/mol increase in ΔG and a decrease in melting temperature of over 40° , claiming this to be in line with their expectations. However, they also show that reversing the polarity of the ion pair within the same scaffold (F49K, F84E) has a 10° lower melting temperature again. The authors attribute this to unfavorable interactions with the helix dipole, but this explanation itself highlights the central problem, as it implies that local factors play a considerable role, even in the conservative case of polarity reversal. For example, charged residues may undergo significant conformation restraint upon burial and have different conformational preferences relative to the residues of the parent they replace. In more practical cases these factors will compound, also extending into the shell of flanking mutations. The authors simply lack the data needed to make any conclusions on such complex systems.

Finally, the authors present a survey of buried ion pairs in membrane proteins, investigating how natural proteins stabilize the buried charges. While this work is well conceived and of considerable interest, the results are analyzed in very broad terms. For example, the distance profiles shown for tyrosine or serine in supplementary figure 4 are considerably different to that for asparagine and glutamine, yet all four are classified as favoured flanking residues. Clearly, much context is being lost in this analysis and it is a stretch to represent this as a "motif" as most readers would understand the term.

I recommend that this work be published in a specialist journal, where it can be configured such that proteins designers and others in the field can get more out of the data.

Reviewer #4 (Remarks to the Author):

The manuscript from Baumgart et al. describe a set of experiments, structures, models, and simulations that aim to address how charged-residue pairs are stabilized within the core of proteins. They find that amphiphilic residues around the charged ones can significantly stabilize the protein. As a test, they designed a protein with two charged pairs of residues and four surrounding glutamines. They found that the crystal structure matched the model of the designed peptide, and that it was extremely resistant to denaturation.

I found this work to be very thorough, and the multiple methods used complemented each other well. As protein design continues to grow, I think this work will be well received by the community.

My only major request is that the authors also investigate the stability of maquette 3 without the glutamines (and/or maybe with just two of them?) to confirm that they are stabilizing the protein as predicted.

Although outside the scope of the present study, I think it would be interesting to look at how these charge pairs affect folding rates/pathways.

Abstract: "how point mutation may alter..." change mutation -> mutations

Page 3 (top line): Remove the comma after "unaffected"

Answer to comments by Reviewer #2

Comment 1: *“This paper describes the introduction of ion-pairs into the hydrophobic core of a designed protein as a model system for protein engineering. Overall the work is sound, and adds significantly to the understanding of the effect of buried polar interactions in folded proteins. The paper lacks clarity at times, especially in the (lack of) description of the rationale for the mutations chosen, as well as in some of the methods employed. I believe that this paper should be accepted pending the following alterations.”*

Answer: We thank this reviewer for the encouraging comments and excellent suggestions that have helped us to further clarify our work.

Specific points:

Question 1: *“The abstract refers to a point mutation- this is misleading given that there are two substitutions.”*

Answer: We have now clarified the plural in the revised abstract:

“Our findings provide a molecular understanding of functional charged networks and how point mutations may alter the protein’s conformational landscape.”

Question 2: *“The introduction of the modifications to Maquette 1 is very vague. No mention is made in the text of the specific modifications to this protein sequence.”*

Answer: The exact modifications are now better clarified in the main text, and the sequences are given in *Extended Data Table 1*. The main text has been changed accordingly:

“To study how the protein stability is affected by buried charged residues, we introduced a glutamate (Glu)/lysine (Lys) ion-pair into the hydrophobic protein core of an artificial 4 α -helical protein scaffold at positions 49 and 84 (F49E and F84K), where helix 1 forms contacts with helices 2 and 3 (see Fig. 1a, Extended Data Methods and Table 1).¹⁰”

Question 3: *“This is clearer for the Maquette 2 modifications, but we still aren't given a clear indication on what has been removed in order to add the ion pair.”*

Answer: The exact modifications are now also clarified for *Maquette 2*. Additions in the main text:

“Insertion of the ion-pair at position 17 and 72 (F17E/F72K relative to Maquette 2/hc) yields a stable structure in the MD simulations (Fig. 1i, Extended Data Fig. 3), but similar to the Maquette 1 model, the charged element forms transient interactions with a few water molecules during the simulations.”

Question 4: *“All protein modifications should be named using the conventional A123B nomenclature, e.g. L50K, where a Leu at position 50 has been changed to a Lys.”*

Answer: We agree that this is a standard convention, which has now been adopted in the revised text. However, as the ion-pairs do not exist in the protein frameworks with the packed hydrophobic core, we also wish to emphasize the ion-pair position itself. We have therefore changed the main text accordingly:

“Insertion of the ion-pair at position 17 and 72 (F17E/F72K relative to Maquette 2/hc)”

“Moreover, the overall protein stability remains unaffected when the charged residues are introduced into the more water accessible region of the bundle (I101E/F50K) (Extended Data Fig. 3c).”

“To this end, we optimised the Glu17-Lys72 ion-pair (F17E/F72K) by introducing random in silico-point mutations in the protein surroundings, and directed the residue search using a Metropolis Monte Carlo (MC)-criterion towards an increased interaction between the ion-pair and the protein structure as an optimisation target.”

“We note that the buried Trp68 might stabilise the Glu17-Lys72 ion-pair in the Maquette 2 model (F17E/F72K, Extended Data Fig. 2c).”

“We find that introduction of Asn at position 69 (Maquette 2/V69N) leads to a small, but statistically significant enhancement of the protein stability in chemical unfolding experiments (Fig. 3f).”

Question 5: “Page 2: 'reverting the polarity of the ion-pair' should be 'inverting the polarity of the ion-pair'”

Answer: We reformulated the sentence:

“inverting the polarity of the ion-pair”

Question 6: “Page 2: A 10 degree C change in melting temperature is not a 'small decrease'. This is actually a very large change, and warrants more explanation, especially given the later discussion of the role of peripheral polar residues in stabilising (or not) buried ion pairs.”

Answer: We thank the reviewer for pointing out this observation, and we agree that the effect is indeed larger than expected. We therefore re-measured the data, and found an inconsistency in the previous buffer conditions and experimental fits for this variant. Based on the new experiments and fits, the two constructs show only $\Delta\Delta T_m=0.1^\circ\text{C}$, which we consider more realistic. We have reformulated this section and revised the manuscript and Supporting Information accordingly, with the new measurements shown below.

“Although helix dipole-sidechain charge interactions may affect the protein stability,¹¹⁻¹³ we find here that inverting the polarity of the ion-pair in the Maquette 1 model only leads to a minor shift in the melting temperature ($\Delta T_m = 0.1^\circ\text{C}$) (Extended Data Figure 1f, 2a).”

13. Sali D, Bycroft M, Fersht A.R. Stabilization of protein structure by interaction of α -helix dipole with a charged side chain. *Nature* **335**, 740–743 (1988).

Extended Data Fig. 1f:

f, temperature melting curves of ip (F49E/F84K, dark blue) and inverted ip (F49K/F84E, light blue). The melting points were obtained from sigmoidal fits, shown as black lines.

Revisions in Extended Data Fig. 2a:

Question 7: “The discussion of the maquette 3 crystal structure states that the structure ‘closely resembles’ the models. This should be quantified, either for the snapshot used in the figure, or more meaningfully, for a minimum energy structure arising from the model building process. Either way this should be quantified by an rmsd value (backbone rmsd makes most sense).”

Answer: We now report that the backbone RMSD between the *in silico* model and the x-ray structure is 1.7 Å. Revisions in the main text:

“The $2F_o - F_c$ electron density map closely resembles the designed *in silico* model (backbone RMSD (*in silico*/x-ray)=1.7 Å), with a charged core forming tight ion-pairs between Lys162/Glu65 and Lys35/Glu136 at distances of 2.7-2.9 Å.”

Answer to comments by Reviewer #3

Comment: *“This paper describes studies on the design of artificial proteins with charged residue pairs within their cores. The study takes a multi-disciplinary approach to evaluating the resulting model proteins, applying molecular dynamics simulations, structural biology and a range of biophysical techniques. As such, the study is well executed and will no doubt be of interest to the protein design community. However, there are major weaknesses, particularly with the selection of the model systems, that seriously undermine the conclusions. As a result, the paper cannot be published in its present form.”*

Answer: We thank this reviewer for the insightful suggestions and for recognizing that our work is well executed and that it will be of interest to the protein design community. In the answers below, we have addressed each comment and revised the manuscript accordingly.

Specific points:

Question 1: *“The authors select three model proteins with which to test mutations that introduce ion pairs into the protein core. All three are four-helical bundles and all three stem from earlier efforts by the groups of Dutton and Moser (authors references 8-10) and DeGrado (reference 13) to design proteins that can bind co-factors within the bundle core, with the aim of supporting oxidoreductase functionalities. This description alone makes the choice of these targets questionable for a study into natural, globular proteins. Little rationale is provided as to why these systems were selected, other than to avoid "the potential bias of natural protein scaffolds". The authors do not clearly establish the relevance of these model systems to proteins such as complex I or Hsp90, which they mention in the introduction”*

Answer: In the revised text, we have now better clarified that the four-helical bundles provide a minimal protein framework that allows us to address the physico-chemical principles of buried ion-pairs. We agree with the reviewer that these proteins do not structurally resemble natural globular proteins or highly intricate proteins such as complex I or Hsp90, which is arguably also why they provide a clean basis to probe the stability of buried ion-pairs without the complicated effects of complex natural surroundings. We have also clarified that from an experimental perspective, the bundles are well-behaved and allow us to deduce interaction principles from rather simple biophysical experiments. Revisions in main text:

*“To address this puzzling question without the potential bias of natural protein scaffolds, we introduce here buried ion-pairs into the hydrophobic core of artificial helical bundle scaffolds^{8,9} by combining directed computational design and molecular simulations with biophysical, spectroscopic, and structural experiments. **These experimentally well-behaved, minimal de novo models do not structurally resemble highly intricate natural proteins, but despite their simple structure, four-helical bundles have been used to probe, e.g., oxidoreductase^{9,10}, oxygen-binding⁸, light-capturing¹⁵, and ion-transport properties¹⁴. Here they have been used to deduce interaction principles of buried ion-pairs without the complicated effect of complex natural surroundings based on rather simple biophysical and computational experiments.**”*

8. Koder, R. L. *et al.* Design and engineering of an O₂ transport protein. *Nature* **458**, 305–309 (2009).
9. Robertson, D. E. *et al.* Design and synthesis of multi-haem proteins. *Nature* **368**, 425–432 (1994).
10. Farid, T. A. *et al.* Elementary tetrahelical protein design for diverse oxidoreductase functions. *Nat. Chem. Biol.* **9**, 826–833 (2013).
14. Joh, N.H. *et al.* *De novo* design of a transmembrane Zn²⁺-transporting four-helix bundle. *Science* **346**, 1520–1524 (2014).
15. Polizzi, N. F. *et al.* *De novo* design of a hyperstable non-natural protein-ligand complex with sub-Å accuracy. *Nat. Chem.* **9**, 1157–1164 (2017).

Question 2: “A case in point is the first example protein, labelled *Maquette 1*. It is based on a parent designed to bind two heme groups in its core via two bis-histidine ligation sites. The protein consists of four identical repeats of 26 residues connected by glycine/serine linkers. These are intended to fold into a four-helical bundle with an up-down topology. Each repeat has a histidine residue in position 6, such that the bis-histidine sites are formed between helices $\alpha 1$ - $\alpha 3$ at one end of the bundle and helices $\alpha 2$ - $\alpha 4$ at the other. Data presented in reference 10 shows nanomolar affinity for heme, but also that the protein forms a molten globule-like state in the absence of the ligand. NMR data present here in supplementary figure 2f are qualitatively in agreement with that presented in reference 10; i.e. the parent protein lacks a well-defined tertiary structure. Nevertheless, the authors introduce an ion pair into this system, via F49E and F84K mutations, with the aim of introducing an $\alpha 2$ - $\alpha 4$ interaction. The ¹⁵N-HSQC spectrum of this *Maquette 1* variant, also shown in supplementary figure 2f does nothing to indicate that this fold has been achieved, and is in no way diagnostic of a “well-folded structure”, as the authors claim.”

Answer: We used here the phenylalanine variant instead of the H6-variant, the sequence of which is now also reported in Extended Table 1. We have now also clarified in the revised manuscript that conclusions about the tertiary structure of *Maquette 1* cannot be drawn from the HSQC spectrum, as this protein is highly symmetric and therefore shows a poorly dispersed NMR spectrum. We observe a similarly dispersed HSQC spectrum for *Maquette 3*, despite our well-resolved x-ray structure. This spectrum is now shown in the Extended Data Figure (see below).

In the revised text, we now also explain that our MD simulations suggest that the protein forms a well-packed core, and our unfolding data show a two-state folding transition, similar to the other protein models, a behaviour that would not be expected for an unstructured protein. We also wish to point out that the energetic effects of introducing the ion-pair for the *Maquette 1* model fits well to our observations for *Maquette 2* and *3*, supporting that our conclusions are robust. Revisions in main text:

“Removal of the glutamate or lysine counter charges decreases the protein stability by around 3 kcal mol⁻¹ based on unfolding experiments (Extended Data Fig. 1, 2a). However, *despite the well-packed protein models in our MD simulations and a two-state folding transition in the chemical unfolding experiments, our nuclear magnetic resonance (NMR) experiments show heteronuclear single quantum coherence (HSQC) spectra that are not too well-dispersed and contain some overlapping signals (Extended Data Fig. 2f), an effect that may arise from the highly symmetric structure of these models.*

Question 3: “Given the quality of the NMR data, it is not surprising that details on this variant are not presented - its structure, internal dynamics or the formation of the ion pair - even unique resonance assignment would prove difficult. The Maquette 1 variant is therefore of no value in supporting the author's claims.”

Answer: As described above, due to the highly symmetric structure of the *Maquette 1* variant, NMR was not the method of choice to address these questions. However, we argue that our combined MD simulations, PBE calculations, CD measurements, and chemical unfolding experiments of *Maquette 1* provide valuable support for the general conclusions of this work. The data provided for *Maquette 1* also support trends observed for *Maquette 2* and *3*, and for natural proteins. We therefore decided to include our design experiments of *Maquette 1* in this manuscript along with these additions in the main text:

“Although we currently lack experimental structural data to draw definite conclusions about the buried ion-pairs in the Maquette 1 models, our data support that buried charges decrease the overall protein stability similarly to what has been described for natural proteins^{4,7} and consistent with the data for other de novo protein models (see below).”

“Similar to the Maquette 1 models, the HSQC spectrum of the Maquette 3 (Extended Data Fig. 6f) shows several overlapping and less sharp signals possibly due to the higher internal symmetry of the model. We could therefore not use NMR to study these models further.”

Question 4: “The third model protein, *Maquette 3*, faces similar issues. No parent sequence is provided for this design (e.g. in supplementary Table 1) and its source is not clearly stated. A BLAST search reveals it to be closely related to a protein deposited in the PDB as 5VJS, a longer four-helical bundle designed to accommodate two heme and a Zn²⁺ binding site. From the little information provided, the ion pair introduced into this parent to form *Maquette 3* does not appear to be inserted into the hydrophobic core, as the reader may expect, but it overlaps with the Zn²⁺ binding site.”

Answer: In addition to the template source that is clarified in the SI-methods section, we have now further clarified the design approach, parent sequence and templates, as well as model sequences in the Supporting Information.

Extended Data Table 1 | Sequences of the *de novo*-protein *Maquettes*.

Construct	Sequence
Maquette 1 Hydrophobic core (hc) <small>Scaffold source Ref. 47</small>	G EIWKQFE DALQKFE EALNQFEDLKQL GSGSGSGG EIWKQFE DALQKFE EALNQFEDLKQL GSGSGSGG EIWKQFE DALQKFE EALNQFEDLKQL GSGSGSGG EIWKQFE DALQKFE EALNQFEDLKQL
Maquette 1 Buried positive charge (F49K)	G EIWKQFE DALQKFE EALNQFEDLKQL GSGSGSGG EIWKQFE DALQK K E EALNQFEDLKQL GSGSGSGG EIWKQFE DALQKFE EALNQFEDLKQL GSGSGSGG EIWKQFE DALQKFE EALNQFEDLKQL
Maquette 1 Buried negative charge (F49E)	G EIWKQFE DALQKFE EALNQFEDLKQL GSGSGSGG EIWKQFE DALQK E E EALNQFEDLKQL GSGSGSGG EIWKQFE DALQKFE EALNQFEDLKQL GSGSGSGG EIWKQFE DALQKFE EALNQFEDLKQL
Maquette 1 Buried ion-pair (1ip/F49E/F84K)	G EIWKQFE DALQKFE EALNQFEDLKQL GSGSGSGG EIWKQFE DALQK E K E EALNQFEDLKQL GSGSGSGG EIWKQFE DALQK K E EALNQFEDLKQL GSGSGSGG EIWKQFE DALQKFE EALNQFEDLKQL

Maquette 1 Buried ion-pair with reversed polarity (1ip_{rev}/F49K/F84E)	G	EIWKQFE	DALQKFE	EALNQFEDLKQL	GGSGSGSGG	
		EIWKQFE	DALQK KE	EALNQFEDLKQL	GGSGSGSGG	
		EIWKQFE	DALQK EE	EALNQFEDLKQL	GGSGSGSGG	
		EIWKQFE	DALQKFE	EALNQFEDLKQL		
Maquette 2 Hydrophobic core (hc) Scaffold source Ref. 48	S	EFEKLRQ	TGDELVQ	AFQRLREIFDK	GD	
		DDSLEQV	LEEIEEL	IQKHRQLFDNR	QEAA	
		DTEAAKQ	GDQWVQL	QRFREAIK	GD	
		KDSLEQL	LEELEQA	LQKIRELAEKKN		
Maquette 2 Buried ion-pair (1ip/F17E/F72K)	S	EFEKLRQ	TGDELVQ	AE QRLREIFDK	GD	
		DDSLEQV	LEEIEEL	IQKHRQLFDNR	QEAA	
		DTEAAKQ	GDQWVQL	KQR FREAIK	GD	
		KDSLEQL	LEELEQA	LQKIRELAEKKN		
Maquette 2 F17E	S	EFEKLRQ	TGDELVQ	AE QRLREIFDK	GD	
		DDSLEQV	LEEIEEL	IQKHRQLFDNR	QEAA	
		DTEAAKQ	GDQWVQL	KQR FREAIK	GD	
		KDSLEQL	LEELEQA	LQKIRELAEKKN		
Maquette 2 F72K	S	EFEKLRQ	TGDELVQ	AE QRLREIFDK	GD	
		DDSLEQV	LEEIEEL	IQKHRQLFDNR	QEAA	
		DTEAAKQ	GDQWVQL	KQR FREAIK	GD	
		KDSLEQL	LEELEQA	LQKIRELAEKKN		
Maquette 2 Charge stabilized ion-pair (1ip/V69N)	S	EFEKLRQ	TGDELVQ	AE QRLREIFDK	GD	
		DDSLEQV	LEEIEEL	IQKHRQLFDNR	QEAA	
		DTEAAKQ	GDQW N QL	KQR FREAIK	GD	
		KDSLEQL	LEELEQA	LQKIRELAEKKN		
Maquette 2 Charge stabilized ion-pair (1ip/L43Y)	S	EFEKLRQ	TGDELVQ	AE QRLREIFDK	GD	
		DDSLEQV	LEEIEEL	Y QKHRQLFDNR	QEAA	
		DTEAAKQ	GDQWVQL	KQR FREAIK	GD	
		KDSLEQL	LEELEQA	LQKIRELAEKKN		
Maquette 2 Charge stabilized ion-pair (1ip/L39Q)	S	EFEKLRQ	TGDELVQ	AE QRLREIFDK	GD	
		DDSLEQV	LEE Q EEL	IQKHRQLFDNR	QEAA	
		DTEAAKQ	GDQWVQL	KQR FREAIK	GD	
		KDSLEQL	LEELEQA	LQKIRELAEKKN		
Maquette 2 Charge stabilized ion-pair (1ip/L94N)	S	EFEKLRQ	TGDELVQ	AE QRLREIFDK	GD	
		DDSLEQV	LEEIEEL	IQKHRQLFDNR	QEAA	
		DTEAAKQ	GDQWVQL	KQR FREAIK	GD	
		KDSLEQL	LEE NE QA	LQKIRELAEKKN		
Maquette 2 Charge stabilized ion-pair (1ip/L91H)	S	EFEKLRQ	TGDELVQ	AE QRLREIFDK	GD	
		DDSLEQV	LEEIEEL	IQKHRQLFDNR	QEAA	
		DTEAAKQ	GDQWVQL	KQR FREAIK	GD	
		KDSLEQL	H EELEQA	LQKIRELAEKKN		
Maquette 2 Charge stabilized ion-pair (1ip/R76S)	S	EFEKLRQ	TGDELVQ	AE QRLREIFDK	GD	
		DDSLEQV	LEEIEEL	IQKHRQLFDNR	QEAA	
		DTEAAKQ	GDQWVQL	KQR F SEAIK	GD	
		KDSLEQL	LEELEQA	LQKIRELAEKKN		
Maquette 2 hydrophilic ion-pair (F50K/I101E)	S	EFEKLRQ	TGDELVQ	AFQRLREIFDK	GD	
		DDSLEQV	LEEIEEL	IQKHRQL K DNR	QEAA	
		DTEAAKQ	GDQWVQL	QRFREAIK	GD	
		KDSLEQL	LEELEQA	LQ KE RELAEKKN		
SVJT		GSPELRQ	EHQQLAQ	EFQQLLQEIQQQ	GRELLKG	ELQGIKQ
		PEKKS ^{SVL}	QKILEDE	EKHIELLETLQQ	TGQEAQQ	LLQELQQ
		PELRQKH	QQLAQKI	QQLLQKHQQQ	LGA	KILEDEE
		DELRELL	KGELQGI	KQYRELQQLGQK	AQQLVQK	LQQTGQK
					LWQLG	LWQLG
Maquette 3 Stabilized charged cluster (2ip/4Q)	M	ASPELRQ	EFQQLIQ	EFQQLLQEIQQQ	IRELLKI	KLQ IIKQ
		PEKKS ^{SVL}	Q KQ LELE	E KQ IELLETLQQ	TAQEAQQ	LLQELQQ
		PELRQKF	QQLAQKI	QQLLQKFQQLVA	KQ LEDEE	KFIELLE
		DELRELL	KG K LQVI	K Q QRELLQLVQK	AQQLVQK	LQQTGQK
					LW	LW
Maquette 3 Stabilized charged cluster (2ip/Q61A/Q68A)	M	ASPELRQ	EFQQLIQ	EFQQLLQEIQQQ	IRELLKI	KLQ IIKQ
		PEKKS ^{SVL}	Q K ALELE	E K AIELLETLQQ	TAQEAQQ	LLQELQQ
		PELRQKF	QQLAQKI	QQLLQKFQQLVA	KQ LEDEE	KFIELLE
		DELRELL	KG K LQVI	K Q QRELLQLVQK	AQQLVQK	LQQTGQK
					LW	LW

47. Farid, T. A. *et al.* Elementary tetrahelical protein design for diverse oxidoreductase functions. *Nat. Chem. Biol.* **9**, 826–833 (2013).

48. Polizzi, N. F. *et al.* De novo design of a hyperstable non-natural protein-ligand complex with sub-Å accuracy. *Nat. Chem.* **9**, 1157–1164 (2017).

Revisions in Methods section:

The *Maquette 3* model was built *in silico* based on a helical coiled-coil template (PDB ID: 5VJT). First, all prosthetic groups were removed from the template, and glycine and alanine residues were placed by valine and isoleucine residues, respectively, to stepwise maximise the non-polar packing, followed by a short (20 ns) MD simulation to test its stability. To further maximise the non-polar packing, hydrophobic residues at positions H10F, H111F, H125F, H139F, and the charged network was modelled manually by substitutions E162K, E35K, I61Q, H68Q, I132Q, Y169Q according to the charge-shielding principles described in Fig. 3e. The final sequence of the *Maquette 3* model is shown in Extended Data Table 1.

Question 5: “*The authors do demonstrate that this design has been successful with the crystal structure of a variant where the ion pair has been stabilized by four flanking glutamine residues, which is clearly a publishable result. But these flanking residues do not seem to solely replace hydrophobic residues, but also tyrosine and histidine residues. In the authors' own survey, tyrosine and histidine were identified as more favored flanking residues than the glutamines that replace them (supplementary Fig. 4).*”

Answer: We thank this reviewer for the comment. We have now clarified in the revised manuscript that tyrosine and histidine residues were indeed represented in the statistical analysis, but were unfortunately too bulky and hindered the protein packing when we tried to model them around the Glu-Lys ion-pairs that possibly provide additional constraints within the protein. We therefore chose the motif consisting of four glutamine residues. We have further clarified that the glutamines fitted well into the packed core, and provided an additional dynamic flexibility within the hydrogen-bonded networks of the charge shielding motif. Revision in the main text:

“Although other charge-stabilising residues, such as tyrosine and histidine, are often present around natural ion-pairs (Extended Data Fig. 4, 5), these residues were too bulky to be inserted around the charged network in the current minimal Maquette 3 model.”

Question 6: “*This again undermines the main claims of the paper. As no data is presented on the parent, we cannot know how much stabilization has been achieved by the ion pair and flanking mutations, also bearing in mind that the parent design (5VJS) was conceived for functionally binding multiple cofactors, not for stability.*”

Answer: To address this excellent question, we now built a new *Maquette 3* model with only two glutamines surrounding the charged residues (substitutions Q61A and Q68A), and probed the stability by chemical unfolding experiments. We observe a 5 kcal mol⁻¹ decrease in the overall protein stability, supporting that these residues indeed provide a charge-shielding effect for the buried ion-pair. Revisions in main text:

“Nevertheless, to quantify the stabilising effect of the charge shielding residues surrounding the ion pairs, we further substituted two of the glutamines with alanine residues (Q61A, Q68A). Our chemical unfolding measurements suggest that these substitutions indeed destabilise the protein by around 5 kcal mol⁻¹, supporting the proposed charge-shielding effect (Extended Data Fig. 6g).”

g. *Maquette 3* with 2ip and Q61A/Q68A, CD spectrum, chemical unfolding profile with GdnHCl up to 8M, and ΔG value calculated from chemical unfolding experiments. Data points are presented as mean values of triplicates with \pm standard deviation as error bars. The error of the ΔG values was propagated from the standard deviation of the mean chemical unfolding profiles during the fit.

Question 7: “As no NMR data is presented on this set of variants, we cannot correlate the prevalence of the ion pair with that reported for *Maquette 2*.”

Answer: We have now measured an HSQC spectrum of *Maquette 3*, shown in Extended Data Fig. S6f, but as expected due to its more symmetric structure, the spectrum is unfortunately not of the same quality as for the *Maquette 2* models. This prevents us from using NMR to further explore its biophysical properties. Instead, our high-resolution x-ray diffraction data illustrate the structure and charge-shielding principles of this protein, whilst chemical unfolding experiments were used to probe its thermodynamic stability, and atomistic molecular dynamics simulations were used to probe the dynamics of the system. Revisions in main text:

”Similar to the *Maquette 1* models, the HSQC spectrum of the *Maquette 3* (Extended Data Fig. 6f) shows several overlapping and less sharp signals possibly due to the higher internal symmetry of the model. We could therefore not use NMR to study these models further.”

Extended Data Fig. 6f:

Extended Data Fig. 5 | **f**, HSQC of 2ip/4Q *Maquette 3*.

Question 8: “*This leaves us with the example of Maquette 2.*”

Answer: As explained above, we argue that NMR, despite its beauty and usefulness to probe the *Maquette 2* models, provides less information for the symmetrical *Maquette 1* and *3* models. However, other complementary methods do provide insightful information that corroborate that -the buried ion-pairs in all our protein variants have similar biophysical properties. We therefore consider that *Maquettes 1* and *3* also provide valuable insight into the buried ion-pairs.

Question 9: “*Here the parent is the PSI protein designed by DeGrado (author's reference 13). This follows a similar strategy to the parent proteins of Maquette 1 in that it incorporates a co-factor binding site - in this case for a porphyrin - within a four-helical structure, but aims to support this site with a clearly separated hydrophobic core region. The authors exploit this separation to place an ion pair within the hydrophobic region and seek to stabilize this with various flanking mutations. They then solve the structure of the ion-pair variant by NMR. Although this provides a good example of the applicability of NMR in such cases, there are several questions with the results. Firstly, as the authors mention, the 15N-HSQC fingerprint of the ion-pair variant is surprisingly different to that of the parent protein, indicative of structural changes on a global scale. From Fig. 2c, the overall dispersion of signals for the variant appears smaller than that of the parent and the linewidths of the variant appear larger. It is thus reasonable for the authors to conclude that the variant undergoes considerable internal motion. However, they do not quantify and localize these dynamic changes, for example via NMR relaxation studies. Given the importance of internal dynamics to their argument, this is a puzzling omission.*”

Answer: To address this excellent question, we performed NMR relaxation studies of the *Maquette 2* models, by conducting hetNOE, T_1 , and T_2 relaxation experiments, that are now shown in Extended Data Fig. 8. All constructs showed similar signals, suggesting that the substitutions do not drastically change the overall dynamics at least on the ps-ns timescale, that all models remain folded to a similar extent under the studied conditions, and that the overall structure is not significantly influenced by the introduction of the ion-pair within the hydrophobic core. Although solving the NMR-structure of *Maquette 2 ip/N69* is outside the scope of the present work, our relaxation data suggest that its structure is overall similar to that of the *Maquette 2/ip* construct. Revisions in the main text:

“To further investigate the conformational flexibility of the Maquette 2 models, we performed heteronuclear NOE (hetNOE), as well as longitudinal (T_1) and transverse (T_2) relaxation experiments.¹⁸ The hetNOE and relaxation data of the hydrophobic core, ion-pair, and charge-stabilised ion-pair constructs are very similar (Extended Data Fig. 8), indicating that all models have similar dynamics on the ps-ns timescale, maintain a similar fold under the studied conditions, and are likely to have overall similar structures.”

Extended Data Fig. 8:

Extended Data Fig. 8 | Relaxation data of *Maquette 2* models. The figure shows hetNOE, T_1 , and T_2 relaxation data for **a**, hydrophobic core, **b**, ion-pair, and **c**, charged-stabilised ion pair *Maquette 2*.

Addition in the Methods section:

^{15}N T_1 , $T_{1\rho}$, and steady-state heteronuclear NOE experiments were collected at an external magnetic field of 11.74 T (^1H Larmor frequency of 500 MHz) and 293 K. Temperature compensation was employed, according to Lakomek *et al.*³⁶ The ^{15}N rf amplitude for the $T_{1\rho}$ spin-lock was set to 1.5 kHz. For the steady-state heteronuclear NOE experiment a saturation time of 4 s was employed. T_2 was calculated according to

$$\frac{1}{T_2} = \frac{R_{1\rho}}{\sin^2 \theta} - \frac{R_1}{\tan^2 \theta} \text{ with } R_{1\rho} = 1/T_{1\rho}, R_1 = 1/T_1 \text{ and } \tan \theta = \frac{\omega_1}{\Omega}.$$

ω_1 is the amplitude of the ^{15}N spin-lock field and Ω is the ^{15}N resonance offset from the spin-lock carrier frequency. The experimental error was set to two times the standard deviation of the spectral noise. Uncertainties were estimated by 1000 Monte Carlo runs.

36. Lakomek, N. A., Ying J., Bax A. Measurement of ^{15}N relaxation rates in perdeuterated proteins by TROSY-based methods. *J. Biomol. NMR* **53**, 209-221 (2012).

Question 10: “Equally puzzling is the lack of a structure on the V69N stabilized variant. If the flanking mutation is stabilizing the ion pair and decreasing the extent of internal motion, then this should be evident in the ^{15}N -HSQC spectrum, which should at least partially return to the dispersion and linewidths of the parent. Such data are, however, not provided, nor is any on the five further flanking mutations that were either neutral or mildly destabilizing, according to supplemental Fig. 2d.”

Answer: We have now measured and assigned the HSQC spectrum for the *Maquette 2/V69N* variant. A shift of some peaks can indeed be seen due to the introduced mutation, with a general

downfield shift, and we note that the dispersion is very similar to that of the hc *Maquette 2* model. However, solving the structure of *Maquette 2/V69N* is unfortunately outside the scope of the present work. Additions in the main text and SI:

“The HSQC spectrum of Maquette 2/V69N indicates that the protein is well-folded, with a downfield shift of most peaks relative to the ion-pair model (Extended Data Fig. 2g).”

Extended Data Fig. 2 | g, HSQC of ip/V69N and ip *Maquette 2*.

Question 11: *“Taken together, these issues mean that there is little consistent data that the authors can use to support their general hypothesis. For example, they show that introducing an F49E, F84K ion pair into Maquette 1 results in a ~5 kcal/mol increase in ΔG and a decrease in melting temperature of over 40°, claiming this to be in line with their expectations. However, they also show that reversing the polarity of the ion pair within the same scaffold (F49K, F84E) has a 10° lower melting temperature again. The authors attribute this to unfavorable interactions with the helix dipole, but this explanation itself highlights the central problem, as it implies that local factors play a considerable role, even in the conservative case of polarity reversal. For example, charged residues may undergo significant conformation restraint upon burial and have different conformational preferences relative to the residues of the parent they replace. In more practical cases these factors will compound, also extending into the shell of flanking mutations. The authors simply lack the data needed to make any conclusions on such complex systems.”*

Answer: The general destabilising effects of buried ion-pairs in the *Maquette 1* models are consistent with our estimates for the other bundles based on the chemical unfolding experiments. We agree with the reviewer that the buried ion-pairs might also affect the protein stability by introducing strain and/or changes in conformational entropy, but probing such effects is unfortunately outside the scope of the present work.

We thank the reviewer for raising the 10°C-shifting in melting temperature, and we agree that the effect is indeed larger than expected. We therefore re-measured the data, and found an inconsistency in the previous buffer conditions and experimental fits for this variant. Based on the new experiments and fits, the two constructs show only a $\Delta\Delta T_m=0.1^\circ\text{C}$, which we consider more realistic. We have reformulated this section and revised the manuscript accordingly.

“Although helix dipole-sidechain charge interactions may affect the protein stability,¹¹⁻¹³ we find here that inverting the polarity of the ion-pair in the Maquette 1 model only leads to a minor shift in the melting temperature ($\Delta\Delta T_m=0.1$ °C) (Extended Data Figure 1f, 2a).”

13. Sali D, Bycroft M, Fersht A.R. Stabilization of protein structure by interaction of α -helix dipole with a charged side chain. *Nature* **335**, 740–743 (1988).

“Although we currently lack experimental structural data to draw definite conclusions about the buried ion-pairs in the Maquette 1 models, our data support that buried charges decrease the overall protein stability similar to what has been described for natural proteins^{4,7} and consistent with the results of our de novo protein models (see below). To improve the accuracy of the design, ”

Question 12: *“Finally, the authors present a survey of buried ion pairs in membrane proteins, investigating how natural proteins stabilize the buried charges. While this work is well conceived and of considerable interest, the results are analyzed in very broad terms. For example, the distance profiles shown for tyrosine or serine in supplementary figure 4 are considerably different to that for asparagine and glutamine, yet all four are classified as favoured flanking residues. Clearly, much context is being lost in this analysis and it is a stretch to represent this as a "motif" as most readers would understand the term.”*

Answer: We thank the reviewer for this excellent suggestion. A possible reason for the high occurrence of tyrosine/serine residues could arise from their overall higher abundance within the dataset. We find in general that charged/polar residues are under-represented, whilst nonpolar and bulky residues are found more often, as expected for membrane proteins. However, tyrosine residues, somewhat diverge from this trend by being over-represented within the dataset. A possible explanation could be that the tyrosine residues comprise both bulky non-polar and aromatic properties that can be used for packing the protein core, but also a polar hydroxy headgroup that can form hydrogen-bonds, in addition to its π -cation forming properties with positively charged residues. We have now also clarified by further analysing the distance distributions, that arginine, glutamate/aspartate, and tyrosine residues are observed more often within the proximity of ion-pairs. Interestingly, glutamine residues seem to favour Arg-Asp ion-pairs, whilst Lys-Glu ion-pairs favour asparagines, possibly due to steric constraints. Probing such details of the character of ion-pairs and surrounding residues will be addressed in future studies. Revisions in main text:

“Interestingly, tyrosine residues that have an overall slightly higher natural abundance, are somewhat over-represented within the membrane proteins dataset (Extended Data Fig. 5b), possibly as they comprise both bulky non-polar/aromatic properties that can be used for packing the protein core or form cation- π interactions with positively charged residues, but also a polar hydroxy headgroup that can form hydrogen-bonds.”

”The ion-pair distributions further suggest that glutamine residues seem to favour Arg-Asp ion-pairs, whilst Lys-Glu ion-pairs favour asparagine residues (Extended Data Fig. 5a), possibly due to steric constraints in natural proteins.”

Extended Data Fig. 5 | Amino acid occurrence depending on the ion-pair composition and natural abundance of amino acids in the OPM data set. a, Natural occurrence of amino acids around different types of ion-pairs as a function of distance relative to the ion-pair **b**, Natural abundance of amino acids as compared to their abundance within the OPM dataset. Amino acid abundances of proteins within the OPM dataset are shown as dots with the average marked as grey lines, whereas individual natural amino acid abundances⁴⁵ are shown as black lines. For each residue, the outliers, $1N$, with values above 0.4 are not shown.

45. Lukasz P. Kozlowski, Proteome-pI: proteome isoelectric point database, *Nucleic Acids Research*, **45**, D1112–D1116 (2017).

Answer to comments by Reviewer #4

Comment 1: “The manuscript from Baumgart et al. describe a set of experiments, structures, models, and simulations that aim to address how charged-residue pairs are stabilized within the core of proteins. They find that amphiphilic residues around the charged ones can significantly stabilize the protein. As a test, they designed a protein with two charged pairs of residues and four surrounding glutamines. They found that the crystal structure matched the model of the designed peptide, and that it was extremely resistant to denaturation.”

“I found this work to be very thorough, and the multiple methods used complemented each other well. As protein design continues to grow, I think this work will be well received by the community.”

Answer: We thank this reviewer for finding our work thorough and interesting for the field, and for the comments that have helped us to further improve our work.

Question 1: “My only major request is that the authors also investigate the stability of maquette 3 without the glutamines (and/or maybe with just two of them?) to confirm that they are stabilizing the protein as predicted.”

Answer: To address this excellent question, we now built a new *Maquette 3* model with only two glutamines surrounding the charged residues (substitutions Q61A and Q68A), and probed the stability by chemical unfolding experiments. We observe a 5 kcal mol⁻¹ decrease in the overall protein stability, supporting that these residues indeed provide a charge-shielding effect for the buried ion-pair.

Revisions in main text: “to quantify the stabilising effect of the charge shielding residues surrounding the ion-pairs, we further substituted two of the glutamines with alanine residues (Q61A, Q68A). Our chemical unfolding measurements suggest that these substitutions indeed destabilise the protein by around 5 kcal mol⁻¹, supporting the proposed charge-shielding effect (Extended Data Fig. 6g).”

g, *Maquette 3* with 2ip and Q61A/Q68A, CD spectrum, chemical unfolding profile with GdnHCl up to 8M, and ΔG value calculated from chemical unfolding experiments. Data points are presented as mean values of triplicates with \pm standard deviation as error bars. The error of the ΔG values was propagated from the standard deviation of the mean chemical unfolding profiles during the fit.

Question 2: “Although outside the scope of the present study, I think it would be interesting to look at how these charge pairs affect folding rates/pathways.”

Answer: We fully agree with the reviewer on this excellent suggestion, and can happily confirm that such studies are in progress, and will follow after publication of our current work.

Question 3: “Abstract: “how point mutation may alter...” change mutation -> mutations
Page 3 (top line): Remove the comma after “unaffected””

Answer: We have revised the sentences accordingly:

“Our findings provide a molecular understanding of functional charged networks and how point mutations may alter the protein’s conformational landscape.”

“Moreover, the overall protein stability *remains unaffected* when the charged residues are introduced...”

REVIEWER COMMENTS

Reviewer #2 (Remarks to the Author):

I am satisfied that the issues raised in the initial review have been dealt with, and the manuscript should now be published.

Reviewer #3 (Remarks to the Author):

Due to technical problems, I will not be able to submit a detailed review for this paper. From the first round of reports, I also see that the other reviewers support publication. Although I still think this paper should be published elsewhere to provide a level of detail not possible in the present format, I will therefore likewise support publication after some important corrections.

Maquette1

The authors speculate that the poor spectra of the maquette 1 series could be due to the internal symmetry of the proteins. I have considerable experience with NMR of repeat proteins, and this type of broadened, low-dispersion spectra are not characteristic. Qualitatively, the spectra for maquette 3 are very different and in no way support this conclusion. A molten globule like or partially aggregated state for maquette 1 and its variants is a far more likely explanation.

The corrected melting temperature for the polarity-reversed variant does now make more sense, but also highlights how weak this data really is. If variations in melting temperature of +/- 10 C can be argued away by invoking helix dipole interactions, then what is this data telling us?

Maquette 2

There must be some error in the new HSQC overlay for the ion-pair and glutamine variants. The best explanation for the considerable uniform shift in the proton dimension is a change in the shift of water due to different measurement conditions, e.g. temperature. This must be checked and corrected. The analysis of this data and the new relaxation data is unfortunately too limited to add any much-needed depth to the paper.

Maquette 3

The reader is still left to find out for themselves that the parent of maquette 3 is a designed zinc binding protein, where two of the coordinating glutamates have been mutated to lysine. Note that I do not find this approach problematic per se, but the reader needs to know what has been done and why. As the parent is published as a raw PDB entry without an associated paper, background on its properties relative to maquette 3 is almost non-existent. Do we even know its melting temperature? What was its parent sequence?

There are many issues with the relevance of these examples to natural proteins, as I mentioned in my first review. For one, just consider that designed proteins have sequences that are optimized for a specific fold and function and therefore may have a far more rugged sequence landscape than a natural equivalent. Mutations around the design may then simply be exploring the design principals used to create the parent. Once again, I am not opposed to this approach, but such issues warrant much more space than the authors have here.

Finally, note that there will not only be biases due to residue frequencies in the authors' survey of membrane proteins, but also spacial biases, as some residues - e.g. tyrosine - have preferred positions relative to the membrane. Without careful calibration, such a survey risks recapitulating well known principles of protein chemistry.

Reviewer #4 (Remarks to the Author):

The authors have thoroughly addressed the reviewers' concerns.

Answer to REVIEWER COMMENTS

Reviewer #2 (Remarks to the Author):

I am satisfied that the issues raised in the initial review have been dealt with, and the manuscript should now be published.

Answer: We thank this reviewer for supporting publication.

Reviewer #3 (Remarks to the Author):

Due to technical problems, I will not be able to submit a detailed review for this paper. From the first round of reports, I also see that the other reviewers support publication. Although I still think this paper should be published elsewhere to provide a level of detail not possible in the present format, I will therefore likewise support publication after some important corrections.

Answer: We thank the reviewer for the encouraging comments and that he/she now also supports publication after revisions. We agree that our work raises also many new detailed questions, which could be of interest for a specialized audience, and can be addressed in future work.

Comment 1: Marquette1

The authors speculate that the poor spectra of the marquette 1 series could be due to the internal symmetry of the proteins. I have considerable experience with NMR of repeat proteins, and this type of broadened, low-dispersion spectra are not characteristic. Qualitatively, the spectra for maquette 3 are very different and in no way support this conclusion. A molten globule like or partially aggregated state for maquette 1 and its variants is a far more likely explanation.

Answer: We appreciate the suggestions of this reviewer. In our interpretation, the high symmetry of this model is affecting the number of peaks and overlapping signals, but we recognize the possibility that, *e.g.*, molten-globule like states could also be involved. We have therefore revised the discussion to include this possibility:

"our nuclear magnetic resonance (NMR) experiments show heteronuclear single quantum coherence (HSQC) spectra that are not too well-dispersed and contain some overlapping signals (Extended Data Fig. 2f), an effect that may arise from the highly symmetric structure of these models, **and/or involvement of possible molten-globule like states.**¹⁰"

Question 2: The corrected melting temperature for the polarity-reversed variant does now make more sense, but also highlights how weak this data really is. If variations in melting temperature of +/- 10 C can be argued away by invoking helix dipole interactions, then what is this data telling us?

Answer: As described in the text, the focus is here on the introduction of the ion-pair in the hydrophobic protein core that shows a clear thermodynamic effect in chemical unfolding experiments, and is supported across the series of different models.

Question 3: Maquette 2

There must be some error in the new HSQC overlay for the ion-pair and glutamine variants. The best explanation for the considerable uniform shift in the proton dimension is a change in the shift of water due to different measurement conditions, e.g. temperature. This must be checked and corrected. The analysis of this data and the new relaxation data is unfortunately too limited to add any much-needed depth to the paper.

Answer: We had unfortunately added a wrong version of this figure, with a mistake in the referencing, during the final stages of the revisions. We thank the reviewer for pointing this out, and apologize for the mistake. We agree with the reviewer that our systems contain interesting data that can be addressed in future work. The correct panel is shown below, and the revised figure has been added in the Supporting Information.

Questions 4: Maquette 3

The reader is still left to find out for themselves that the parent of maquette 3 is a designed zinc binding protein, where two of the coordinating glutamates have been mutated to lysine. Note that I do not find this approach problematic per se, but the reader needs to know what has been done and why. As the parent is published as a raw PDB entry without an associated paper, background on its properties relative to maquette 3 is almost non-existent. Do we even know its melting temperature? What was its parent sequence?

Answer: We have now clarified in the main text that the scaffold was initially designed to bind metals. We have not studied the properties of the parent protein model further, as the target was focusing on creating a cluster of buried ion-pairs. Additions in the main text:

"ca. 70 Å-long α -helical Maquette-protein framework (Maquette 3, Fig. 4c, d, see Extended Methods), originally a Zn²⁺-site binding scaffold (PDB ID: 5VJT), that provides a larger non-polar interaction surface relative to the minimal 4 α -helical Maquette models 1 and 2."

Question 5: There are many issues with the relevance of these examples to natural proteins, as I mentioned in my first review. For one, just consider that designed proteins have sequences that are optimized for a specific fold and function and therefore may have a far more rugged sequence landscape than a natural equivalent. Mutations around the design may then simply be exploring the design principals used to create the parent. Once again, I am not opposed to this approach, but such issues warrant much more space than the authors have here.

Answer: Our work finds similar trends for introducing buried ion-pairs in three independent constructs that were explored here, suggesting that the effects do not arise from design bias. A more detailed study that could probe how different substitutions may generally affect the stability in these artificial bundle proteins could be interesting, but is outside the scope of the present work.

Question 6: Finally, note that there will not only be biases due to residue frequencies in the authors' survey of membrane proteins, but also spacial biases, as some residues - e.g. tyrosine - have preferred positions relative to the membrane. Without careful calibration, such a survey risks recapitulating well known principles of protein chemistry.

We have clarified in the revised manuscript that some amino acids can have special preferred positions:

"We note, however, that certain amino acids may have special preferred locations in the protein structure.¹⁶"

¹⁶Hessa T. *et al.* Molecular code for transmembrane-helix recognition by the Sec61 translocon. *Nature* **450**, 1026-1030 (2007).

Reviewer #4 (Remarks to the Author):

The authors have thoroughly addressed the reviewers' concerns.

Answer: We thank this reviewer for supporting publication.